# Redox-switchable breathing behavior in tetrathiafulvalene-based metal–organic frameworks

Jian Su[1], Shuai Yuan[2], Hai-Ying Wang[1], Lan Huang [3], Jing-Yuan Ge[1], Elizabeth Joseph[2], Junsheng Qin[2], Tahir Cagin[2,4], Jing-Lin Zuo [1] & Hong-Cai Zhou [2,3]

Metal–organic frameworks (MOFs) that respond to external stimuli such as guest molecules, temperature, or redox conditions are highly desirable. Herein, we coupled redox-switchable properties with breathing behavior induced by guest molecules in a single framework. Guided by topology, two flexible isomeric MOFs, compounds **1** and **2**, with a formula of $In(Me_2NH_2)$ (TTFTB), were constructed via a combination of $[In(COO)_4]^-$ metal nodes and tetratopic tetrathiafulvalene-based linkers (TTFTB). The two compounds show different breathing behaviors upon the introduction of $N_2$. Single-crystal X-ray diffraction, accompanied by molecular simulations, reveals that the breathing mechanism of **1** involves the bending of metal–ligand bonds and the sliding of interpenetrated frameworks, while **2** undergoes simple distortion of linkers. Reversible oxidation and reduction of TTF moieties changes the linker flexibility, which in turn switches the breathing behavior of **2**. The redox-switchable breathing behavior can potentially be applied to the design of stimuli-responsive MOFs.

[1] State Key Laboratory of Coordination Chemistry, School of Chemistry and Chemical Engineering, Collaborative Innovation Center of Advanced Microstructures Nanjing University, Nanjing 210093, China. [2] Department of Chemistry, Texas A&M University, College Station, TX 77843–3255, USA. [3] Department of Materials Science and Engineering, Texas A&M University, College Station, TX 77843-3003, USA. [4] Artie McFerrin Department of Chemical Engineering, Texas A&M University, College Station, TX 77843-3022, USA. Jian Su and Shuai Yuan contributed equally to this work. Correspondence and requests for materials should be addressed to J.-L.Z. (email: zuojl@nju.edu.cn) or to H.-C.Z. (email: zhou@chem.tamu.edu)

Metal–organic frameworks (MOFs) are a promising class of highly ordered porous materials with diverse applications in the fields of gas storage and separation, sensing, and catalysis[1–5]. Over the past few decades, a significant number of MOFs with various structures, porosities, and framework compositions have been extensively explored[6]. Amongst them, flexible MOFs have garnered particular interest because they combine crystalline order of the underlying coordination framework with cooperative structural transformability[7–12]. Additionally, they are able to respond to various chemical and physical stimuli such as light, pressure, temperature, redox environment, or the introduction of guest molecules[13–18]. A representative example is the breathing effect in which the framework experiences a reversible unit-cell dimensional change resulting from guest adsorption or desorption[19]. This leads to unique sorption behaviors which have never been observed in other systems[20–22]. On the other hand, the redox activity of MOFs has been explored through the introduction of redox-active metal nodes or organic ligands. These materials have shown promise for applications in microporous conductors, electrocatalysts, energy storage devices, and electrochemical sensors[23–27].

The host structures in flexible MOFs can be altered by external stimuli, potentially altering the availability of the cavity to guest molecules. Building off this concept, we intended to couple the breathing behavior induced by guest molecules with redox-switchable properties in a single framework. The resulting materials are expected to show dual-stimuli-responsive behavior. Compared with other porous materials, the crystalline nature of MOFs provides a unique advantage through the study of their behavior via crystallography, therefore maximizing the understanding of the correlation between applied stimuli and ensuing properties[28–30]. To realize redox regulation of MOF porosity, a flexible framework with redox-active triggers is indispensable. The tetrathiafulvalene (TTF) is a sulfur-rich conjugated molecule with two reversible and easily accessible oxidation states (i.e., radical TTF$^{\bullet+}$ cation and TTF$^{2+}$ dication) that have been widely studied as a critical electron donor component for conductive[25–27], optoelectronic, and magnetic[23,24] materials. The oxidation of TTF to TTF$^{\bullet+}$ or TTF$^{2+}$ converts the $7\pi$-electron dithiolylidene ring to the aromatic $6\pi$-electron configuration, and consequently results in a rigid, fully conjugated aromatic system. Bearing this in mind, the TTF moiety was incorporated into the linker of a MOF as a redox switch to control the flexibility of the framework. Although numerous flexible MOFs have been reported, redox-controlled flexibility in MOFs has never been studied.

Herein, two flexible isomeric MOFs with TTF moieties, compounds **1** and **2**, were constructed using topology analysis as a guide. The two compounds display different breathing behaviors derived from the bending of linkers, the distortion of metal–ligand bonds and the relative sliding of interpenetrated networks. The TTF moieties allow for the reversible oxidation by $I_2$ and reduction via treatment with *N,N*-dimethylformamide (DMF), which switches the linker flexibility and the breathing behavior of compound **2**. As compound **2** is an anionic MOF, the effect of counterions on the breathing behavior is also studied using cation exchange experiments.

## Results

**Topology-guided design of flexible MOFs**. To obtain a flexible MOF with redox-responsive behavior, a topology-guided design was adopted. The tetratopic TTF-based linker, tetrathiafulvalene tetrabenzoate (TTFTB), was adopted as the redox switch, which can be regarded as a 4-connected square planar node (Fig. 1b). Topologically, the 4-connected square planar linker is able to

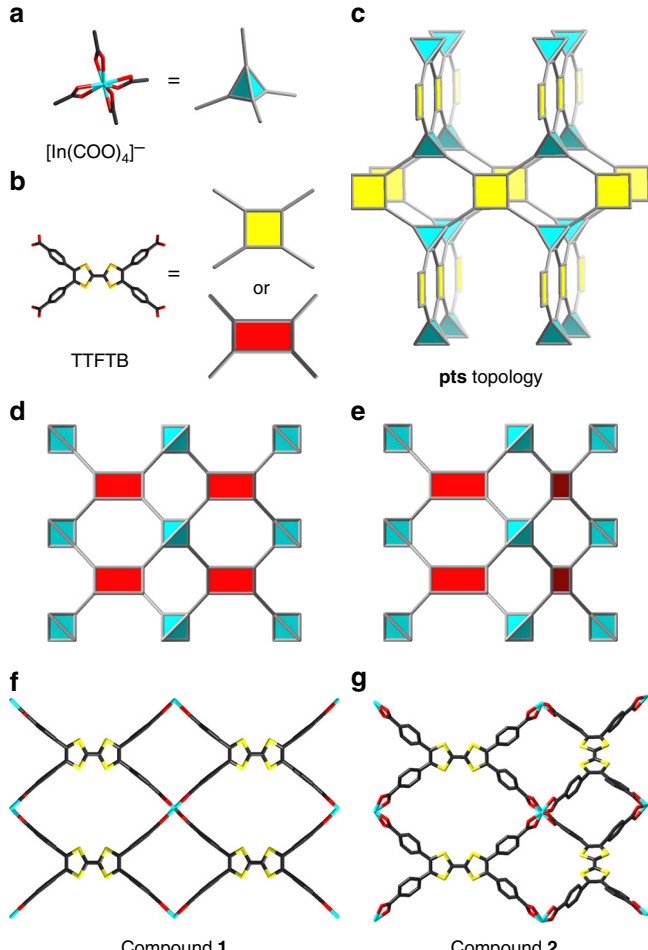

**Fig. 1** Topology-guided design of compounds **1** and **2**. **a** [In(COO)$_4$]$^-$ inorganic building unit as a tetrahedral node; **b** TTFTB organic linker as a square planar node; **c pts** topology; **d, e** two networks derived from **pts** topology; **f, g** structure of compounds **1** and **2**. Color scheme: In, cyan; O, red; C, gray; S, yellow

form a variety of binodal networks when combined with different metal nodes. Among the numerous MOFs with square planar linkers in the literature, flexible frameworks have rarely been observed. The reported MOFs with square planar linkers are generally formed by inorganic building units that have high connectivity, mainly resulting in rigid frameworks. In this respect, the [In(COO)$_4$]$^-$ seems to be a suitable inorganic building unit for the construction of flexible MOFs as it is a mononuclear metal node with relatively low connectivity. Topologically, it can be simplified into a 4-connected tetrahedral vertex (Fig. 1a), which could be extended by combination with square planar linkers into a **pts** type framework (Fig. 1c). The **pts** net is expected to show flexibility by changing bond angles between nodes.

As expected, the reaction between In$^{3+}$ and TTFTB gave rise to the designed MOF structure. Interestingly, two isomeric MOFs were isolated by tuning the synthetic conditions. The isomeric MOFs displayed the same topology but different arrangements of linkers. Although the TTFTB was topologically simplified into a square, it shows a rectangular geometry in the structure. Taking this into consideration, two structures can be derived from the **pts** net. If all the linkers are aligned with the same orientation throughout the structure, compound **1** will be generated (Fig. 1d, f). On the other hand, if linkers with perpendicular

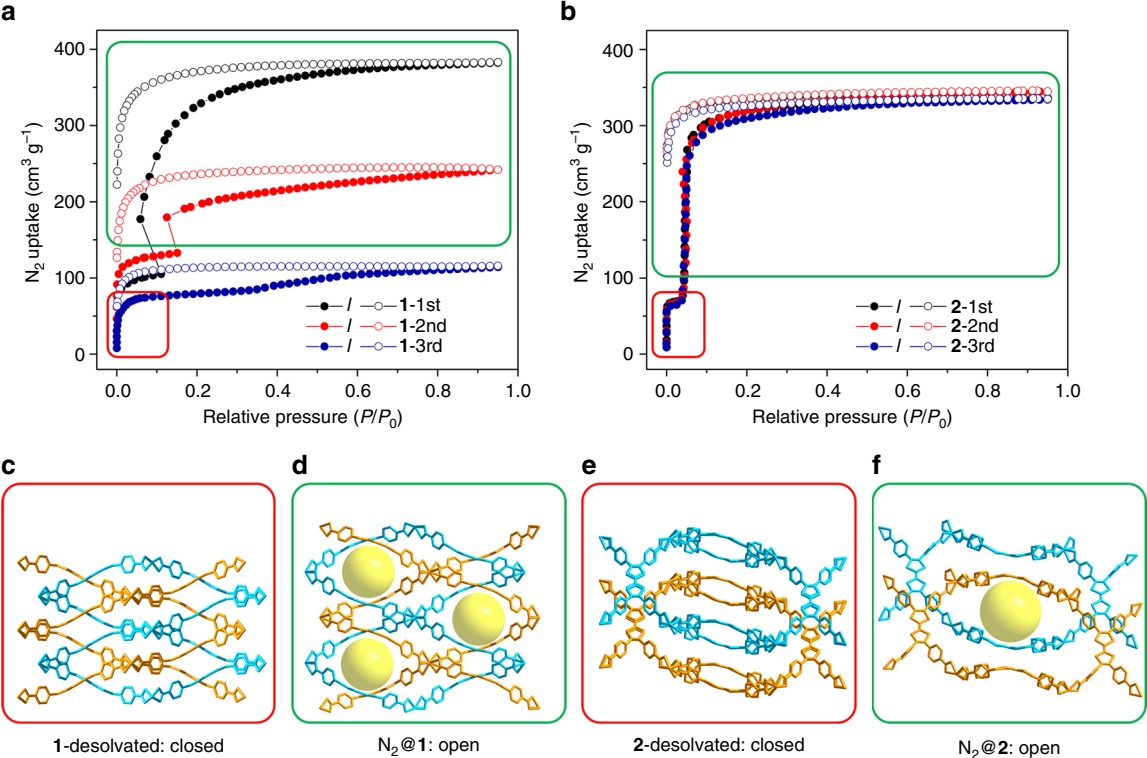

**Fig. 2** Breathing behaviors of compounds **1** and **2**. **a**, **b** $N_2$ sorption isotherms of compounds **1** and **2** for three adsorption–desorption cycles; **c**, **d** simulated structures of **1** upon activation and $N_2$ adsorption; **e**, **f** single-crystal structure of **2** upon activation and $N_2$ adsorption. Two sets of interpenetrated networks were colored blue and orange. The cavities are highlighted by yellow spheres

orientations are arranged alternately within the structure, compound **2** (Fig. 1e, g) will be observed.

Single-crystal X-ray diffraction (SC-XRD) analysis reveals that compound **1** crystallizes in the orthorhombic space group *Cccm* (Supplementary Tables 1 and 2). The asymmetric unit (Supplementary Fig. 1a) contains half of an $In^{3+}$ ion, half of a TTFTB ligand, and half of a dimethylammonium cation ($Me_2NH_2^+$). As shown in Supplementary Fig. 1b, the $[In(COO)_4]^-$ adopts a distorted tetrahedron coordination geometry with the four carboxylates from TTFTB ligands in chelating bidentate coordination. The central C=C bond of the TTF unit is 1.343(9) Å, which is comparable of the neutral TTF moiety state. As an anionic framework formed by $[In(COO)_4]^-$ nodes, $Me_2NH_2^+$ exists in the cavity to balance the charge. Overall, compound **1** is a twofold interpenetrated framework with a porosity of 48.2% determined by the PLATON[31] calculations (Supplementary Fig. 2).

Interestingly, a framework isomer, compound **2**, was isolated by reducing the reaction temperature. The crystal structure of compound **2** crystallizes in the triclinic space group $P\bar{1}$ (Supplementary Tables 4 and 5). The asymmetric unit (Supplementary Fig. 3a) contains two $In^{3+}$ ions, two TTFTB ligands, and two molecules of $Me_2NH_2^+$. Two TTFTB with different conformations were observed within the structure (Supplementary Fig. 3b). They are perpendicular to each other and arranged alternately throughout the framework (Supplementary Fig. 4). This arrangement is different from that of the TTFTB in compound **1**, which shows the same configuration and orientation throughout the structure. Compound **2** is also a twofold interpenetrated anionic MOF with $Me_2NH_2^+$ located in the cavity. The PLATON[31] calculations show that the total solvent-accessible volume gives a porosity of 46.9%. Both compounds **1** and **2** are formulated as $In(Me_2NH_2)(TTFTB)$.

**Breathing behaviors induced by $N_2$ sorption**. Usually, the breathing behaviors of MOFs can be triggered by the adsorption or desorption of guest molecules. Herein, the $N_2$ sorption measurements at 77 K were conducted to assess the porosity and flexibility of compounds **1** and **2**. The $N_2$ uptake of compound **1** quickly reaches 100 $cm^3\,g^{-1}$ at 0.05 bar before saturating. However, with the increase of pressure, another adsorption stage emerges at 0.18 bar which eventually pushes the $N_2$ uptake to 375 $cm^3\,g^{-1}$ (Fig. 2a). Similar to **1**, the $N_2$ isotherm of **2** exhibits low initial $N_2$ uptake (70 $cm^3\,g^{-1}$), followed by another distinct adsorption step (Fig. 2b). However, in **2**, the second adsorption step appears at a lower pressure (0.05 bar). An adsorption vs. desorption hysteresis was observed which is an indication of discontinuous structural transition[32]. The gas adsorption isotherms of compounds **1** and **2** do not fit in any of six isotherm types classified by International Union of Pure and Applied Chemistry. In fact, the stepped isotherms are an indication of flexible frameworks which possess narrow-pore structures when no guests are present, and transform to an open structure triggered by adsorbed guests. Note that the dynamic behavior of compound **1** is not fully reversible, with a loss of crystallinity and porosity after each adsorption–desorption cycle, as suggested by PXRD and $N_2$ isotherms (Supplementary Figs. 5 and 6). On the other hand, the breathing effect of compound **2** is fully reversible for at least three adsorption–desorption cycles.

The most distinctive advantage of MOFs over other porous materials is their long-range ordered crystalline structure, which can provide a unique insight into the structure–property correlation by means of crystallography[29, 33]. Therefore, SC-XRD analysis was carried out to provide direct structural evidence of the breathing behavior, and to shed light on the mechanism. The crystals of compounds **1** and **2** were examined under $N_2$ flow at various temperatures, mimicking the $N_2$

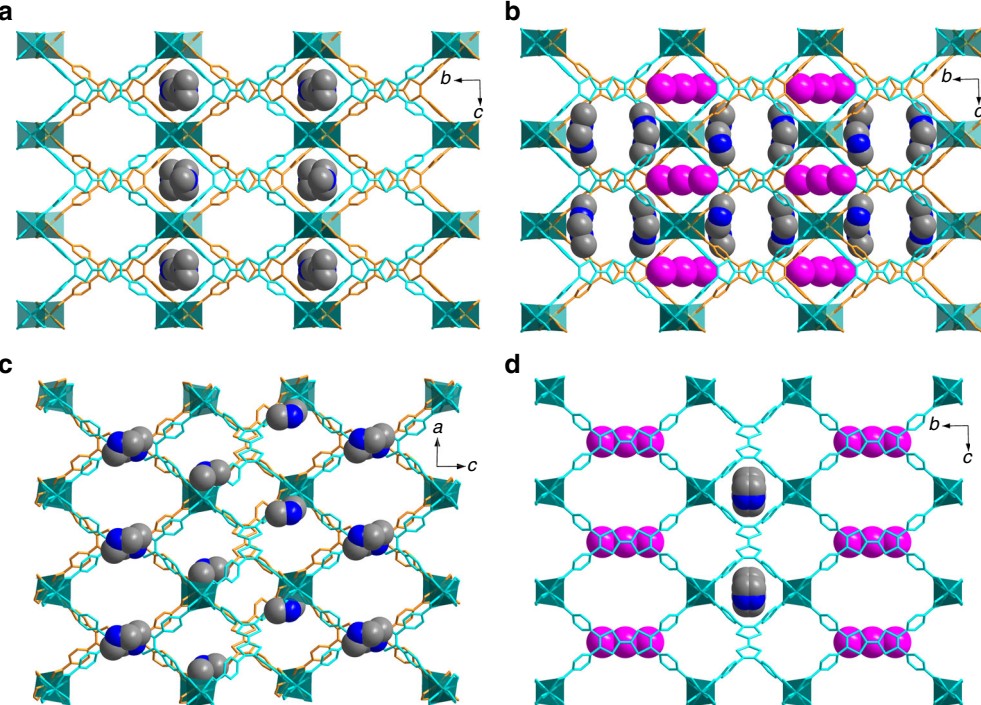

**Fig. 3** Crystal structure changes upon $I_2$ treatment. **a**, **b** Structural transformations of **1** and $I_3^-$@**1**; **c**, **d** guest location in the single-crystal structures of **2**, and $I_3^-$@**2**. Two sets of interpenetrated networks were colored blue and orange. Color scheme of guest species: I, purple; N, blue; C, gray

sorption process (Supplementary Table 8). The desolvated compound **2** shows a narrow-pore structure with blocked channels. The TTF moieties on the linker do not form a fully conjugated aromatic system and the linker adopts a bent conformation. Linkers bent in different directions are alternately arranged within each framework, resulting in a blocked channel and a narrow-pore structure (Fig. 2e and Supplementary Fig. 7). Reducing the temperature to 80 K saturates the pore with $N_2$ molecules, which triggers the pore opening. In the structure of $N_2$@**2**, linkers are arranged along the same direction within the framework so that an open-pore structure is observed (Fig. 2f and Supplementary Fig. 7). In a word, the flexible TTFTB within compound **2** partially changes its conformation, which explains the pore-opening upon $N_2$ uptake.

The diffraction quality of crystals for compound **1** after desolvation is not good enough to allow for structural refinement. Thus, we turned to molecular simulations to explain the breathing behavior. The molecular model of desolvated compound **1** was built from the single structure of the synthesized sample by removing solvent molecules and performing geometry optimization. Compared with the as-synthesized structure, the optimized **1**-desolvated structure shows a shrinkage in unit-cell volume and a mutual movement of two sets of frameworks (Fig. 2c). The simulated total $N_2$ uptake of this narrow-pore structure is 130 $cm^3 g^{-1}$, which matches well with the experimental results. Molecular dynamics simulation reveals that the structure adapts to the loading of $N_2$ molecules through the bending of metal–ligand bonds and the sliding of interpenetrated frameworks (Fig. 2d). The structural change opens the cavity and increases the pore volume, which explains the experimentally observed stepped adsorption isotherms. The change of unit-cell volume and the relative movement of two sets of frameworks in **1** causes a significant structural change which gradually decomposes the framework after each adsorption–desorption cycle. On the other hand, compound **2** undergoes a relatively mild linker flip, resulting in the observation of fully reversible breathing behavior.

**Redox-controlled breathing behaviors.** Experimentally obtained single-crystal structure indicates that the breathing behavior of compound **2** is derived from the flexibility of the TTFTB linker (Fig. 3). Therefore, we hypothesize that the breathing behavior of compound **2** can be controlled by tuning the linker flexibility. It is well known that TTF is an electron donor, and each dithiolylidene ring is a $7\pi$-electron system with two electrons from each S atom and one from each $sp^2$-C atom. Therefore, the TTF is not a fully conjugated aromatic system, which is also reflected by the bent conformation in the crystal structure. However, oxidation of TTF converts the S-heterocycles into an aromatic $6\pi$-electron configuration, consequently transforming the central double bond to essentially a single bond. A conjugated aromatic system is formed which rigidifies the TTF moiety (Fig. 4c). In other words, the oxidation of TTF is expected to turn off its flexibility by forming a fully conjugated aromatic system.

Bearing this in mind, we explored the possibility of switching the MOF flexibility by redox chemistry. Solid state electrochemistry was performed on the compounds **1** and **2** to probe its redox behavior. Cyclic voltammetry of compounds **1** and **2** show the expected TTF processes. The two one-electron oxidation waves were observed at $E_{1/2} = 0.16$ and $0.43$ V vs. Fc/Fc$^+$ for compound **1** and $E_{1/2} = 0.05$ and $0.52$ V vs. Fc/Fc$^+$ for compound **2**, respectively. These can be assigned to the TTF/TTF$^{\bullet+}$ and TTF$^{\bullet+}$/TTF$^{2+}$ redox couples (Supplementary Figs. 9 and 10). $I_2$ was reported as a modest oxidant for conversion of TTF into its radical state TTF$^{\bullet+}$ while maintaining the crystallinity of MOF to the maximum extent. Compounds **1** and **2** were oxidized by soaking their crystals in a solution of $I_2$ in cyclohexane (0.1 M) at room temperature for 12 h. An obvious color change was observed which is attributed to the incorporation of $I_2$ and the formation of TTF$^{\bullet+}$ (Supplementary Fig. 11). The diffuse reflectance ultraviolet–visible–near infrared (UV–Vis–NIR) spectrum of **1** (Supplementary Fig. 12) reveals low intensity bands at 4200 and 13,700 $cm^{-1}$, ruling out the existence of TTF$^{\bullet+}$ in the as-synthesized samples. Upon oxidation with $I_2$, the aforementioned

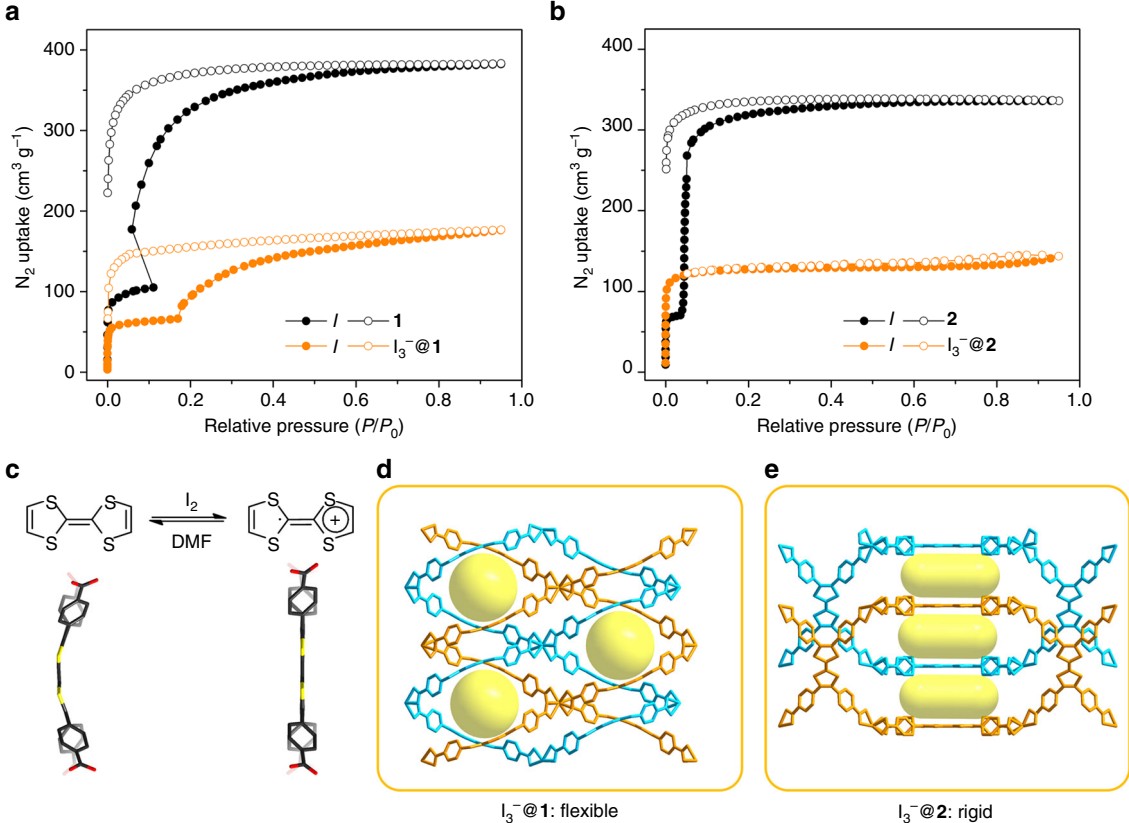

**Fig. 4** Redox-switchable breathing behaviors. **a**, **b** $N_2$ sorption isotherms of **1**, $I_3^-$@**1**, **2**, and $I_3^-$@**2**; **c** schematic representation of the electron configuration and confirmation change upon oxidation/reduction; **d**, **e** single-crystal structures of $I_3^-$@**1** and $I_3^-$@**2**. Two sets of interpenetrated networks were colored blue and orange. The cavities are highlighted by yellow spheres. Color scheme of linkers: O, red; C, gray; S, yellow

bands significantly increase in intensity, which is consistent with the oxidation of neutral TTF to TTF$^{•+}$. This result was corroborated by solid state Vis–NIR spectroelectrochemical measurements (Supplementary Fig. 13), where in situ oxidation of the parent compound over a potential range of 0–2 V led to an intensification of the band at 13,700 cm$^{-1}$. This oxidation process was shown to be reversible because the band at 13,700 cm$^{-1}$ decreased in intensity as the potential was returned to 0 V. Solid state UV–Vis–NIR spectroscopy of **2** revealed weak bands which can be assigned to the TTF radical cation; bands found at 4380 and 13,870 cm$^{-1}$ may be assigned to the [TTF$_2$]$^{•+}$ mixed-valence state and [TTF$_2$]$^{2+}$ π-radical dimer, respectively (Supplementary Fig. 12). The presence of the TTF radical cation in the parent framework **2** is likely due to the low oxidation potential of the initial one-electron oxidation of the TTF moiety (confirmed by electrochemistry of the framework). The spectrum of the iodine-doped species, $I_3^-$@**2**, revealed a significant intensification of the previously described features as a result of the chemical oxidation of the framework with $I_2$.

Electron paramagnetic resonance (EPR) studies of the solid compounds at 110 K further confirmed the generation of TTF$^{•+}$ radical (Supplementary Fig. 14). All EPR spectra (which were recorded under the same conditions) showed an axial set of g values ($g_{x,y}$ = 2.006, $g_z$ = 2.013 for $I_3^-$@**1** and $g_{x,y}$ = 2.005, $g_z$ = 2.010 for $I_3^-$@**2**), typical of TTF$^{•+}$ cation radicals, confirming that the TTF moiety in these compounds is the redox-active unit[23, 34, 35]. A negligible signal in the EPR spectrum of pristine **1** and **2** indicates the presence of the vast majority of the TTF moiety in its neutral form, which is consistent with the very weak band at ~13,500 cm$^{-1}$ in the UV–Vis–NIR spectra[23, 34, 35]. The EPR

signal increased after $I_2$ doping, confirming the increased concentration of the TTF$^{•+}$ radical cation.

The magnetic susceptibility data of polycrystalline samples of $I_3^-$@**1** and $I_3^-$@**2** are displayed in Supplementary Fig. 15 as the $\chi_m T$ vs. T plot in a DC field of 1 kOe. The $\chi_m T$ (0.375 and 0.361 cm$^3$ K mol$^{-1}$ for $I_3^-$@**1** and $I_3^-$@**2** at 300 K, respectively) gradually decreased when the temperature dropped, revealing the existence of antiferromagnetic interactions. The result indicates the magnetic contribution of the radical electrons. The expected $\chi_m T$ value for the isolated $S$ = 1/2 radical electrons (TTF$^{•+}$) was consistent with the result of the EPR measurement. The room temperature electrical conductivities of **1** and **2** (1.23 × 10$^{-9}$ and 1.16 × 10$^{-9}$ S cm$^{-1}$ for **1** and **2**, respectively, Supplementary Fig. 16 and Supplementary Table 7) were significantly improved upon oxidation (5.50 × 10$^{-8}$ and 1.68 × 10$^{-7}$ S cm$^{-1}$ for $I_3^-$@**1** and $I_3^-$@**2**, respectively, Supplementary Fig. 16 and Supplementary Table 7). However, the lack of suitable conducting channels still results in poor conductivity. The altered conductivity can be attributed to the TTF radical cation of the doped compound compared to their neutral counterparts.

Direct structural evidence of $I_3^-$@**1** and $I_3^-$@**2** are provided by X-ray crystallography (Supplementary Tables 3 and 6). Due to their good chemical stability, compounds **1** and **2** maintained single crystalline after the oxidation, thus their structural transformations can be monitored by SC-XRD. The existence of $I_3^-$ as the product of oxidation reaction is observed in the crystal structure of both **1** and **2**, which is relatively rare among $I_2$ doped MOFs (Fig. 3). The oxidized MOFs, namely $I_3^-$@**1** and $I_3^-$@**2**, are isomers with the same formula of In$_2$(Me$_2$NH$_2$)(TTFTB$^{•+}$)$_2$(I$_3^-$). After $I_2$ oxidation, the space group and cell parameters of

compound **1** are very similar (Fig. 4d). However, $I_3^-$ as the product of oxidation reaction was observed in the channel along with a slight conformational change of linkers (Fig. 3b and Supplementary Fig. 17). The central C···C distance of the TTF units increases from 1.343(9) to 1.387(6) Å, corresponding to the partial conversion of neutral TTF to its radical cationic state TTF$^{\bullet+}$. The crystal structure of compound **2**, however, undergoes dramatic change upon $I_2$ oxidation (Fig. 4e). The crystal system transformed from triclinic to monoclinic along with the space group change from $P\bar{1}$ to $P2/m$. The bent TTFTB ligand in compound **2** became coplanar in $I_3^-$@**2**, along with the changes of bond lengths and angles around the $[In(COO)_4]^-$ node (Fig. 3d and Supplementary Fig. 18). The conformational change of TTFTB increases the symmetry by creating an inversion center and a twofold axis through the molecule, which in turn changes the space group of the whole structure. The central C···C distance of the TTF units in $I_3^-$@**2** were measured to be 1.418(13) and 1.416(11) Å, which is significantly longer than the neutral compounds. More importantly, the flexible TTFTB became rigid. Since the breathing effect of **2** is resulted from the conformational change of flexible TTFTB, a rigid linker is expected to result in the loss of breathing behavior (Supplementary Fig. 19).

As expected, the $I_3^-$@**2** shows a type I isotherm indicating a rigid microporous structure (Fig. 4b). The step at 0.05 bar disappeared upon oxidation, suggesting that the framework flexibility is turned off by oxidation. However, the $I_3^-$@**1** still shows a stepped isotherm with hysteresis loops, although the total uptake is reduced due to the pore space that is partially occupied by $I_3^-$ (Fig. 4a). Obviously, the flexibility of compound **1** is not affected by the change within the linkers. This is in line with the breathing mechanism of **1** which involves the bending of metal–ligand bonds and the sliding of two mutually interpenetrated frameworks. The relative movement of two interpenetrated frameworks is independent of the linker flexibility. Note that the oxidation of TTF moieties in compounds **1** and **2** are reversible. $I_3^-$@**1** and $I_3^-$@**2** could return to their initial structures by DMF treatment which reduced the TTF$^{\bullet+}$ and removed the $I_3^-$ (Supplementary Fig. 20). The restored **1** and **2** displayed the same crystal structure and sorption properties as the pristine samples. These results clearly demonstrates that the dynamic behavior of flexible MOFs is a reversible process that can be switched on and off by redox chemistry. It should be noted that oxidation of **1** and **2** by $I_2$ occurs along with the formation of $I_3^-$ as counterions. We cannot rule out the possible influence of $I_3^-$ as guest species on the breathing behavior of MOFs.

**Effects of counterions**. As an anionic framework, the $[Me_2NH_2]^+$ counterions could also play an important role on the behavior of the framework. Indeed, the counterions serve as a special guest molecule that interacts with the framework though coulombic interactions, which could lead to dramatic differences in gas adsorption capacity[36]. However, it is usually difficult to precisely determine the position of the counterions by X-ray crystallography because they are not bound to the framework. This also hinders the estimation of sorption behaviors by molecular simulation as the counterions can move upon temperature changes or gas adsorption/desorption. In fact, the porosities of **2**-desolvated and $N_2$@**2** were estimated by molecular simulation based on their respective single-crystal structures using $N_2$ as a probe, which indicated similar accessible pore volume. Therefore, **2**-desolvated and $N_2$@**2** are expected to have the same theoretical $N_2$ uptake, contradicting the experimental results. This is tentatively explained by the steric effect of the counterions, which are large enough to block the small channel of **2**-desolvated resulting in an almost non-porous structure. The window size of $N_2$@**2**, on

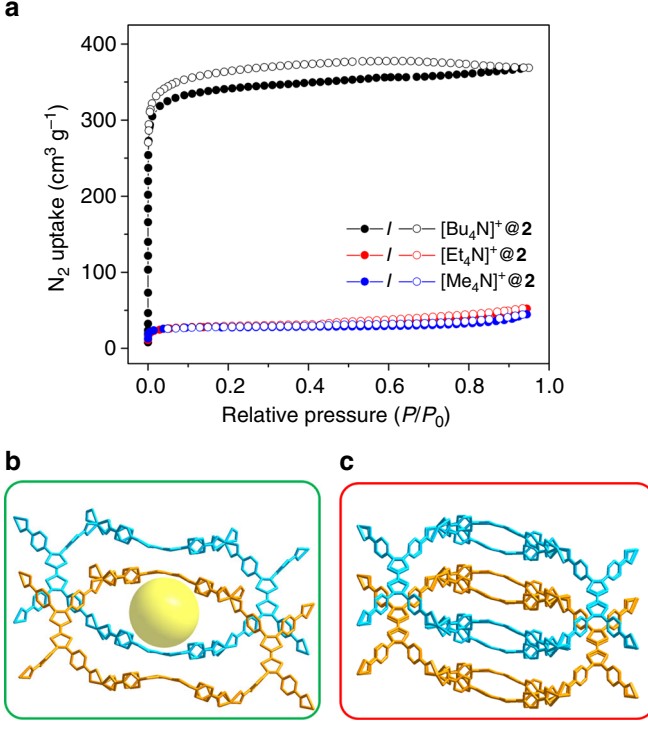

**Fig. 5** Effects of cations. **a** $N_2$ sorption isotherms of $[Bu_4N]^+$@**2**; **b** $N_2$ sorption isotherms of $[Et_4N]^+$@**2**; **c** $N_2$ sorption isotherms of $[Me_4N]^+$@**2**. Two sets of interpenetrated networks were colored blue and orange. The cavity is highlighted by the yellow sphere

the other hand, is large enough for both $[Me_2NH_2]^+$ counterions and $N_2$ molecules resulting in the observation of a porous structure. To systematically investigate the effect of counterions, cationic exchange was conducted on compound **2**.

The $[Me_2NH_2]^+$ can be readily replaced by $[Me_4N]^+$, $[Et_4N]^+$ and $[Bu_4N]^+$ (Supplementary Fig. 22). Based on the [1]H-NMR results (Supplementary Figs. 23–26), $[Me_2NH_2]^+$ was fully replaced by the $[Bu_4N]^+$ cation, whereas the exchange by $Et_4N^+$ was incomplete. $N_2$ adsorption isotherms further demonstrate that $[Bu_4N]^+$@**2** has an open structure without any flexible behavior. On the other hand, the $[Me_4N]^+$@**2** and $[Et_4N]^+$@**2** are almost non-porous (Fig. 5a). This counter-intuitive result is rationalized by the crystal structures of $[Bu_4N]^+$@**2** and $[Et_4N]^+$@**2** (Supplementary Fig. 28 and Supplementary Table 9). Based on the single-crystal structure, the $[Bu_4N]^+$@**2** always shows an open structure (Fig. 5b), while the channels in $[Et_4N]^+$@**2** remain closed even in DMF (Fig. 5c). Therefore, $[Bu_4N]^+$@**2** exhibits a rigid porous structure while $[Et_4N]^+$@**2** is almost non-porous. Note that the linker in the structure of $[Et_4N]^+$@**2** is disordered even at low temperature (100 K), owing to the incomplete cation exchange. In fact, the exchange of $[Me_2NH_2]^+$ by $[Et_4N]^+$ closes the external MOF cavities, which formed a non-porous shell and prevents further ion exchange in the core of the framework. The pore-opening effect of $Bu_4N^+$ is tentatively attributed to its steric hindrance, which prevents the movement of TTFTB linkers. The cation exchange experiment highlights the important role of $[Me_2NH_2]^+$ on the $N_2$-induced structural change of compound **2**.

In conclusion, we show that the breathing behavior of MOFs can be controlled by redox chemistry in a flexible TTF-based MOF. The structural transformations upon guest adsorption and oxidation are investigated by SC-XRD and corroborated by molecular simulations. The effect of counterions on the breathing

behavior are also studied by cation exchange experiments. The redox-controlled dynamic behavior of MOFs is reminiscent of sophisticated biological behavior such as redox regulation of enzymes. We believe that the discovery of redox-switchable breathing MOFs should not only lead to a new generation of unique adsorbents, but also facilitate the applications of flexible MOFs in gas storage and separation, molecular sensing, and switchable catalysis.

## Methods

**Synthesis of compound 1**. To the solution of $H_4TTFTB$ (0.013 g, 0.034 mmol) in 2 mL DMF, a solution of $In(NO_3)_3 \cdot 4.5H_2O$ (0.020 g, 0.052 mmol) in 0.5 mL $H_2O$ was added, followed by the addition of 0.5 mL ethanol and 0.4 mL $CH_3COOH$. The mixture was heated to 120 °C for 72 h, then allowed to cool down to room temperature at the rate of 5 °C h$^{-1}$. The red octahedral crystals (0.018 g) of **1**, [(TTFTB)In]•$(CH_3NH_2CH_3)$•0.7$(C_2H_5OH)$•$(C_3H_7ON)$, were obtained by filtration and washed with DMF and $CH_3COCH_3$ three times, respectively. Thermogravimetric analysis (TGA) data was shown in Supplementary Fig. 8a. Yield 82.57% (based on $H_4TTFTB$). Calcd for $C_{16.4}H_{19.2}InN_2O_{9.7}S_4$ ($M_r = 642.61$): C, 30.65; H, 3.01; N, 4.36%; S, 19.96%. Found: C, 30.20; H, 2.54; N, 2.63%; S, 19.96%. FT-IR (KBr pellet, cm$^{-1}$): 3675 w, 2782 w, 2360 w, 2796 w, 1700 w, 1653 s, 1603 s, 1539 s, 1506 m, 1380 vs., 1174 m, 1137 m, 1097 m, 1018 m, 862 m, 825 m, 765 s, 705 m, 668 m, 569 m, 483 m, 419 m.

**Synthesis of compound $I_3^-$@1**. The iodine doping of $I_3^-$@**1** was undertaken using a diffusion technique. Crystals of **1** were soaked in a solution of iodine in cyclohexane (0.1 M) at room temperature for 12 h. Note that the color of the crystals became deeper as the doping time was increased (Supplementary Fig. 11). The crystals of $I_3^-$@**1**, [(TTFTB)$_2$In$_2$]•$(I_3)$•$(CH_3NH_2CH_3)$•4$(C_6H_{12})$, were washed with cyclohexane. The quantities of iodine incorporated were confirmed by TGA (Supplementary Fig. 21) and elemental analysis: one $I_3^-$ for $I_3^-$@**1** per unit cell. Anal. Calcd. for $C_{46}H_{56}I_3In_2NO_{16}S_8$ ($M_r = 1745.80$): C, 31.65; H, 3.23; N, 0.80; S, 14.69. Found: C, 31.96; H, 1.55; N, 17.23; S, 10.07. Selected IR data (KBr, cm$^{-1}$): 3675 w, 2989 w, 2546 w, 2360 m, 2342 m, 1705 vs., 1653 s, 1604 s, 1526 m, 1405 vs., 1317 m, 1285 s, 1177 m, 1119 m, 1015 s, 864 m, 763 s, 696 m, 668 w, 551 w, 481 w, 418 w.

**Synthesis of compound 2**. To the solution of $H_4TTFTB$ (0.013 g, 0.034 mmol) in 2 mL DMF, a solution of $In(NO_3)_3 \cdot 4.5H_2O$ (0.020 g, 0.052 mmol) in 0.5 mL $H_2O$ was added, followed by the addition of 0.5 mL ethanol and 0.3 mL $CH_3COOH$. The mixture was heated to 60 °C for 72 h. After that, 0.3 mL $CH_3COOH$ was added, and the mixture was heated to 120 °C for 72 h again, then allowed to cool down to room temperature at a rate of 5 °C h$^{-1}$. The red rod-like crystals (0.008 g) of **2**, [(TTFTB)$_2$In$_2$]•$(CH_3NH_2CH_3)_2$•2$(C_2H_5OH)$•2$(C_3H_7ON)$, were obtained by filtration and washed with DMF and $CH_3COCH_3$ three times, respectively. TGA data was shown in Supplementary Fig. 8b. Yield 43.79% (based on $H_4TTFTB$). Calcd for $C_{34}H_{42}In_2N_4O_{20}S_8$ ($M_r = 1312.87$): C,31.30; H, 3.22; N, 4.27%; S, 19.54%. Found: C, 30.20; H, 2.54; N, 2.63%; S, 4.27%. FT-IR (KBr pellet, cm$^{-1}$): 3649 w, 3003 w, b, 2546 w, 1705 s, 1603 s, 1540 vs., 1374 vs., 1175 m, 1137 m, 1100 m, 1016 s, 860 m, 831 s, 783 m, 767 s, 704 m, 674 m, 568 w, 483 m, 436 m.

**Synthesis of compound $I_3^-$@2**. The iodine doping of $I_3^-$@**2** was undertaken using a diffusion technique. Crystals of **2** were soaked in a solution of iodine in cyclohexane (0.1 M) at room temperature for 12 h. Note that the color of the crystals became deeper as the doping time was increased (Supplementary Fig. 11). The crystals of $I_3^-$@**2**, [(TTFTB)$_2$In$_2$]•$(I_3)$•$(CH_3NH_2CH_3)$•4$(C_6H_{12})$, were washed with cyclohexane. The quantities of iodine incorporated were confirmed by TGA (Supplementary Fig. 21) and elemental analysis: one $I_3^-$ for $I_3^-$@**2** per unit cell. Anal. Calcd. for $C_{46}H_{56}I_3In_2NO_{16}S_8$ ($M_r = 1745.80$): C, 31.65; H, 3.23; N, 0.80; S, 14.69. Found: C, 31.96; H, 1.55; N, 17.23; S, 10.07. Selected IR data (KBr, cm$^{-1}$): 3073 w, 2920 m, 1607 m, 1540 m, 1507 m, 1428 s, b, 1283 m, 1176 m, 1138 m, 1102 m, 1016 s, 902 w, 869 m, 856 s, 823 m, 781 m, 769 s, 703 m, 682 m, 485 w, 464 m, 438 m.

**Synthesis of compound [Me$_4$N]$^+$@2, [Et$_4$N]$^+$@2, and [Bu$_4$N]$^+$@2**. The cation exchanged guest@**2** was obtained by the exchange of [Me$_2$NH$_2$]$^+$ counterions. The crystals of compound **2** were immersed in the 0.01 mol L$^{-1}$ [Me$_4$N]Cl, [Et$_4$N]Cl, and [Bu$_4$N]Cl DMF solution, respectively. Over 10 days, the solution was changed every 12 h to facilitate the counterion exchange. After the exchange process was complete, the crystals were immersed in fresh DMF for one day to remove the guest cations. TGA data was shown in Supplementary Fig. 27.

**Characterization**. Elemental analyses for C, H, N, and S were performed on Perkin-Elmer 240C analyzer. FT-IR data were recorded on Vector27 Bruker Spectrophotometer with KBr pellets in the 4000–400 cm$^{-1}$ region. TGA data were obtained on an STA 449C thermal analysis system with a heating rate of 10 °C min

$^{-1}$ under $N_2$ atmosphere. The PXRD were collected with a scan speed of 0.1 s deg$^{-1}$ on a Bruker Advance D8 (40 kV, 40 mA) diffractometer with Cu radiation ($\lambda = 1.54056$ Å) at room temperature. Calculated PXRD patterns were generated using Mercury 3.0. Magnetic susceptibility measurements were performed using a Quantum Design SQUID VSM magnetometer on microcrystalline samples for all compounds. EPR spectra were obtained by using a Bruker EMX-10/12 variable-temperature apparatus at 110 K. Gas sorption measurements were conducted using a Micrometritics ASAP 2020 system. See Supplementary Methods for details.

**Data availability**. The X-ray crystallographic coordinates for structures reported in this article have been deposited at the Cambridge Crystallographic Data Centre (CCDC), under deposition number CCDC 1560234, 1560199, 1560204, 1560221, 1560235, 1560236, 1560237, and 1560977 for compound **1**, $I_3^-$@**1**, **2**, $I_3^-$@**2**, **2-close**, **2-open**, TEA@**2**, and TBA@**2**, respectively. These data can be obtained free of charge from the Cambridge Crystallographic Data Centre via www.ccdc.cam.ac.uk/data_request/cif. All relevant data supporting the findings of this study are available from the corresponding authors on request.

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

## Acknowledgements

This work was supported by the Major State Basic Research Development Program (No. 2013CB922101), and the National Natural Science Foundation of China (Nos. 91433113 and 21631006). The gas adsorption–desorption studies of this research were supported by the Center for Gas Separations Relevant to Clean Energy Technologies, an Energy Frontier Research Center funded by the US Department of Energy, Office of Science, Office of Basic Energy Sciences under Award Number DE-SC0001015. Structural analyses were supported as part of the Hydrogen and Fuel Cell Program under Award Number DE-EE0007049. Prof. Deanna M. D'Alessandro and Dr. Chanel Leong in School of Chemistry, the University of Sydney are acknowledged for their assistance in diffuse reflectance UV–Vis–NIR spectra, solid state electrochemical, and spectroelectrochemical measurements. This computational work was funded by the Robert A. Welch Foundation through a Welch Endowed Chair to H.C.-Z (A-0030). Texas A&M Supercomputing Facility was acknowledged to provide computing resources. The Distinguished Scientist Fellowship Program (DSFP) at KSU is gratefully acknowledged for supporting this work. The authors also acknowledge the financial supports of US Department of Energy Office of Fossil Energy, National Energy Technology Laboratory (DE-FE0026472). S.Y. also acknowledges the Texas A&M Energy Institute Graduate Fellowship Funded by ConocoPhillips and Dow Chemical Graduate Fellowship.

## Author contributions

Original idea was conceived by J.-L.Z. and H.-C.Z.; experiments and data analysis were performed by J.S., S.Y., H.-Y.W., J.-Y.G., and J.Q.; structure characterization was performed by J.S., manuscript was drafted by J.-L.Z. and H.-C.Z. J.S., S.Y., and E.J. All authors have given approval to the manuscript.

## Additional information

**Competing interests:** The authors declare no competing financial interests.

