## [Peer Review File · Nature Communications]

Reviewers' comments:

Reviewer #1 (Remarks to the Author):

This is an interesting manuscript describing the breathing of a TTF containing MOF as a function of redox. Thus, on oxidation with I2 the linker changes from bent to linear on going from TTF to TTF+. In addition, breathing is observed in the two reported compounds 1 and 2 as a function of N2 uptake. This is a comprehensive and detailed study, and one that will be of general interest to the MOF community. The work appears to be well conducted and I would be content to recommend acceptance subject to the following changes.

Page 3: In4+ should be In3+

the nomenclature I2@1 and I2@2 is very misleading, indeed incorrect. These are complexes of I3- (page 12 and 13) not of I2 and they should be represented as such. I would prefer to see the full formulae with a clear indication of what the counter anions/cations are at all times.

Reviewer #2 (Remarks to the Author):

This is interesting work that would merit publication in Nature Communication.

The main comment is the way the topic is introduced. I feel the analogy with enzymes might be appealing on the surface, but is a little too far-fetched for Nature Communication. I suggest the author simply focus on the very interesting possibility to induce changes in the structure due to stimuli. If the authors want to make the case they should add a part demonstrating the analogy rather than mentioning this in vague terms as it is done now.

I am puzzled by the isotherm in 4a; that shows a solid line suggesting that the experiments first increase the pressure, has a jump that decreases the pressure of the reservoir. This is strange and does not make sense to me.

Some minor comments:

- In the abstract it is mentioned compound 1 and 2 without defining what these compounds are; this is not very useful
- Figure 2 c-f: the authors should explain the yellow and blue colors and what the yellow sphere represents in the caption. Also the other figure caption do not define all the features of the figures
- The authors may find A. Ghysels, et al J. Phys. Chem. C 117 (22), 11540 (2013) a useful reference

Reviewer #3 (Remarks to the Author):

This is an interesting paper with an optimistic abstract and an exciting concept of a 'breathing' MOF. The paper is well referenced and with some rather major corrections, it could be re-visited.

Crystallographic comments:

Some confusion with the several CIF files [differently dated] and those received were not consistent with the reported values in the text.

The compound numbering was confused and inconsistent in these CIFs w.r.t the text.

HKL missing from some CIF and in most cases, there was no mention of SQUEEZE and it would thus appear that the PLATON entries were added later.

The CIF for structure I2@1 (CCDC 1560199) shows R1 as 11%, whereas the Table in the paper shows R1 = 8.25%.

A smaller discrepancy in I2@2 between R1 = 11.5% in text and R1 = 11.9% in CIF.

There are several inconsistencies in the temperature of the experiments/results. Of 8 structures,

1, 2 and I2@2 show 296K in the CIF but 153K in the Table.

2-close has 273K in CIF and 296K in Table. TEA@2 has 103K in CIF and 100K in Table. Three others seem to have matching values.

It appears that ISOR has been mis-used (over enthusiastically applied ?) in some structures, effectively suppressing anisotropy, but mostly the refinements can be regarded as sensible, except where the structures contain Iodine. This requires some serious re-evaluation and considered discussion about disorder. The changing state of I2 is crucial to the discussion about redox behaviour and it is not clear from the values given that such an interpretation is correct. They claim I3 anion is 'unambiguously observed in the crystal structure' but with the large residual peaks in the density maps there is room to re-evaluate the occupancy of all atomic sites and to establish firmly the nature of any disorder or partial occupancy of I2.

Response to Referees' Comments:

Reviewer #1:

This is an interesting manuscript describing the breathing of a TTF containing MOF as a function of redox. Thus, on oxidation with I₂ the linker changes from bent to linear on going from TTF to TTF⁺. In addition, breathing is observed in the two reported compounds 1 and 2 as a function of N₂ uptake. This is a comprehensive and detailed study, and one that will be of general interest to the MOF community. The work appears to be well conducted and I would be content to recommend acceptance subject to the following changes.

Response: Thank you for the supportive comments.

1. Page 3: In⁴⁺ should be In³⁺.

Response: Thank you for pointing it out. We have corrected this mistake in the revised manuscript.

The nomenclature I₂@1 and I₂@2 is very misleading, indeed incorrect. These are complexes of I₃⁻ (page 12 and 13) not of I₂ and they should be represented as such. I would prefer to see the full formulae with a clear indication of what the counter anions/cations are at all times.

Response: Thank you for the suggestions. The I₂ treated samples were named as I₃⁻@1 and I₃⁻@2 respectively. They are both formulated as In₂(Me₂NH₂)(TTFTB⁺)₂(I₃⁻), which have been discussed in the revised manuscript. The formulae of compound 1, 2, I₃⁻@1, and I₃⁻@2 have also been added. Since compound 1 and 2 are isomers with the same formula [In(Me₂NH₂)(TTFTB)], they are distinguished by numbering 1 and 2.

Reviewer #2:

This is interesting work that would merit publication in *Nature Communications*.

Response: We appreciate your supportive comments.

1. The main comment is the way the topic is introduced. I feel the analogy with enzymes might be appealing on the surface, but is a little too far-fetched for *Nature Communications*. I suggest the author simply focus on the very interesting possibility to induce changes in the structure due to stimuli. If the authors want to make the case they should add a part demonstrating the analogy rather than mentioning this in vague terms as it is done now.

Response: Thank you for the comments. We have modified the statement of biomimetics in the revised manuscript. We focus on the discussion of stimuli-responsive materials triggered by redox chemistry, instead of bio-inspired materials. The following sentence has been modified in the revised manuscript.

“The redox-switchable breathing behavior can be potentially applied to the design of stimuli-responsive MOFs.”

“Building off this concept, we intended to couple the breathing behavior induced by guest molecules with redox-switchable properties in a single framework. The resulting materials are expected to show dual-stimuli responsive behavior.”

2. I am puzzled by the isotherm in 4a; that shows a solid line suggestion that the experiments first increase the pressure, has a jump that decreases the pressure of the reservoir. This is strange and does not make sense to me.

Response: Yes. The pressure decreased and the nitrogen uptake increased when the structure undergoes a “gate open” during the adsorption. This is attributed to the dosing mode adopted by the instrument. At low pressure range, a certain amount of nitrogen gas was dosed to the tube each time, the pressure was then stabilized and recorded. For a rigid MOF, the pressure and the nitrogen uptake gradually increase with the dosing of nitrogen gas. For a flexible MOF, the initial dosing do not change the structure until the critical pressure was reached ($P/P_0 = 0.11$). Further dosing of nitrogen caused a dramatic structural change which significantly increased the porosity of the material. The “open” phase absorbed much higher amount of nitrogen and therefore reduced the pressure in the sample tube, since the sample tube was sealed during equilibrium process. Even at reduced pressure, the uptake of nitrogen was still much higher than that of the “close” phase. When the data was recorded after equilibrium, the pressure dropped while the nitrogen uptake significantly increased. The material maintained its “open” structure even at reduced pressure as indicated by the desorption curve. Therefore, the isotherms can be explained by significantly increased porosity of MOF upon phase transformation and the dosing mode used by the instrument. The same isotherms were obtained using different instruments including Micromeritics ASAP 2020 and Micromeritics ASAP 2420.

3. In the abstract it is mentioned compound 1 and 2 without defining what these compounds are; this is not very useful.

Response: Thank you for the suggestions. We have defined compound 1 and 2 in the abstract and their formulae were provided in the revised manuscript. The following sentence have been modified in the revised manuscript.

“Guided by topology, two flexible isomeric MOFs, compounds **1** and **2**, with a formula of $\text{In}(\text{Me}_2\text{NH}_2)(\text{TTFTB})$, were constructed *via* combination of $[\text{In}(\text{COO})_4]^-$ metal nodes and tetratopic tetrathiafulvalene-based linkers (TTFTB).”

4. Figure 2 c-f: the authors should explain the yellow and blue colors and what the yellow sphere represents in the caption. Also the other figure caption do not define all the features of the figures

Response: Thank you. Two sets of interpenetrated networks were colored by blue and orange respectively. The cavities are highlighted by yellow spheres. Color scheme: In, cyan; O, red; C, gray; S, yellow. We have explained the color scheme in the caption.

5. The authors may find A. Ghysels, et al J. Phys. Chem. C 117 (22), 11540 (2013) a useful reference

Response: Thank you for your suggestion. The reference has been discussed in the revised manuscript. The following discussions have been added in the revised manuscript.

“An adsorption versus desorption hysteresis was observed which is an indication of discontinuous structural transition.”

Reviewer #3:

This is an interesting paper with an optimistic abstract and an exciting concept of a 'breathing' MOF. The paper is well referenced and with some rather major corrections, it could be re-visited.

Response: We appreciate your comments.

Crystallographic comments:

1. Some confusion with the several CIF files [differently dated] and those received were not consistent with the reported values in the text.

Response: Thank you for pointing it out. The CIF files have been uploaded again, and the reported values in the text have been corrected.

2. The compound numbering was confused and inconsistent in these CIFs w.r.t the text.

Response: Thanks for your suggestion. We have fixed these problems in the updated CIFs, manuscript and supplementary information.

3. HKL missing from some CIF and in most cases, there was no mention of SQUEEZE and it would thus appear that the PLATON entries were added later.

Response: HKL and the PLATON entries have been added into the CIF files according to the suggestions.

4. The CIF for structure I₂@1 (CCDC 1560199) shows R1 as 11%, whereas the Table in the paper shows R1 = 8.25%. A smaller discrepancy in I₂@2 between R1 = 11.5% in text and R1 = 11.9% in CIF.

Response: We have refined the crystal structure of I₂@1 (renamed as I₃⁻@1 in the revised manuscript), and the R1 has been reduced to 4.56%. The crystal data for I₂@2 (renamed as I₃⁻@2 in the revised manuscript) have also been refined to reduce the R1 to 5.48%.

5. There are several inconsistencies in the temperature of the experiments/results. Of 8 structures, 1, 2 and I₂@2 show 296K in the CIF but 153K in the Table. 7. 2-close has 273K in CIF and 296K in Table. TEA@2 has 103K in CIF and 100K in Table. Three others seem to have matching values.

Response: Thanks for pointing it out. We checked all the temperatures, 296K has been changed to 153K in the compounds 1, 2 and I₂@2 (renamed as I₃⁻@1 in the revised manuscript) in the CIF. In 2-close, the value of temperature has been changed to 273K in table. In TEA@2, the value of temperature has been changed to 103K in table.

6. It appears that ISOR has been mis-used (over enthusiastically applied ?) in some structures, effectively suppressing anisotropy, but mostly the refinements can be regarded as sensible, except where the structures contain Iodine. This requires some serious re-evaluation and considered discussion about disorder. The changing state of I₂ is crucial to the discussion about redox behaviour and it is not clear from the values given that such an interpretation is correct. They claim I₃ anion is 'unambiguously observed in the crystal structure' but with the large residual peaks in the density maps there is room to re-evaluate the occupancy of all atomic sites and to establish firmly the nature of any disorder or partial occupancy of I₂.

Response: We have refined crystal structures of 2, I₂@1 and I₂@2 (renamed as I₃⁻@1 and I₃⁻@2 in the revised manuscript). The problems of ISOR and large residual peaks in the density maps have been solved. The residual peaks in the density maps is now reasonable. The I-I bond length of I₃ anion (about 3.0 Å) is larger than I₂ (about 2.7 Å). As shown in I₃⁻@1 and I₃⁻@2, the I-I bond length are 3.0235(8) and 2.9910(9) Å, respectively, corresponding to the I₃⁻ species. Moreover, the TTF unit of I₃⁻@1 and I₃⁻@2 is in its radical TTF•+ cation state with the larger central C···C distance (about 1.40 Å), which requires I₃⁻ anions to balance the charge. The compositions of I₂@1 and I₂@2 were further substantiate by elemental analysis and thermogravimetric analysis. We agree that the statement of ‘unambiguously observed in the crystal structure’ is too absolute, and we have modified these discussions in the revised manuscript.

REVIEWERS' COMMENTS:

Reviewer #2 (Remarks to the Author):

I happy with the revised manuscript and recommend publication as it stands.

Reviewer #3 (Remarks to the Author):

Editorial Note: This Reviewer provided no further comments to the authors.